# Simulation of Four-Point Bending Fracture Test of Steel-Fiber-Reinforced Concrete

**DOI:** 10.3390/ma15207146

**Published:** 2022-10-13

**Authors:** Wenguang Kan, Zailin Yang, Weilong Yin

**Affiliations:** 1College of Aerospace and Civil Engineering, Harbin Engineering University, Harbin 150001, China; 2School of Water Conservancy and Civil Engineering, Northeast Agricultural University, Harbin 150030, China; 3School of Astronautics, Harbin Institute of Technology, Harbin 150001, China

**Keywords:** numerical simulation, flexural strength, fracture content, concrete, steel fiber

## Abstract

To investigate the influence of addition amount and length of steel fibers on the bearing capacity of a concrete beam, this study simulated the crack propagation process of a concrete beam in a four-point bending experiment. The extended finite element method (XFEM) using the ABAQUS software was adopted. Additionally, stress distribution trends for the concrete under loading and load–displacement curves at the stressed points were obtained. The simulation results for a concrete beam with different amounts and lengths of steel fibers were compared and analyzed, and conclusions were drawn. The experiment shows that the flexural performance of the concrete improves with increases in the length and amount of steel fibers, but the reinforcement effects produced by different amounts and lengths of steel fibers are different. When the steel fiber content is 1.5% and the length is 20–25 mm, the reinforcement effect in the concrete is significantly improved, and its flexural strength is nearly doubled.

## 1. Introduction

Concrete material has many advantages, such as easy preparation, low price, high compressive strength, good durability, and so on, so it has been generally used in the construction industry. In recent years, higher requirements for building materials have been proposed, and the disadvantages of plain concrete, such as low tensile strength, easy cracking, and low toughness, have also received significant attention and research [1]. In particular, many efforts have been made to investigate the concrete cracking behavior both in the presence of Fiber-Reinforced-Plastic (FRP) reinforcement [2,3], and induced by steel corrosion [4]. A traditional analytical model, named a Two-Parameter Model [5], for Mode I fracture toughness determination has recently been extended to compute Mixed Mode fracture toughness of cementitious materials [6,7], which has been proved to be a simple tool providing accurate results. It could be also able to achieve the phenomenon of size-effect independence of fracture toughness [8].

Furthermore, a refined diffuse cohesive approach for the failure analysis of quasibrittle materials, such as concrete, has recently been proposed [9,10]. Steel-fiber-reinforced concrete (SFRC) is a type of composite material in which steel fibers with high tensile strength are added to improve crack resistance and bearing capacity. Generally, the cracking of steel-fiber-reinforced concrete is not caused by tensile fracture of steel fibers, but by insufficient bond strength between steel fibers and concrete. The optimal steel fiber amount and length not only obtain higher-strength materials, but also save on materials costs.

Belytsckho and Black [11] proposed an extended finite element method (XFEM), which inherits advantages of the traditional finite element method and has the advantage of solving discontinuous mechanical problems. It has been rapidly and widely used to investigate concrete fracture mechanisms [11,12,13,14,15]. With respect to the traditional finite element method, the XFEM has the advantage of solving discontinuous mechanics. After simplifying the extended finite element theory, ABAQUS introduced the function XFEM since its Version 6.4 [16].

Scientists have carried out several experimental studies on the mechanical properties of fiber-reinforced concretes. Li et al. [17] studied the mechanical properties of steel-fiber-reinforced concrete (SFRC) under freeze–thaw conditions and concluded that with an increase in freeze–thaw cycles, the mechanical properties of plain concrete and steel-fiber-reinforced concrete degraded gradually. However, the mechanical properties of steel-fiber-reinforced concrete were still better than those of plain concrete. Yan et al. [18] studied the mechanical properties of steel-fiber-reinforced concrete at high temperatures, and concluded that the tensile, compressive, and splitting strength of steel-fiber-reinforced concrete are higher than those of plain concrete. Zhao et al. [19] studied the mechanics of steel-fiber-reinforced concrete after corrosion and proposed that with the corrosion of the steel fibers, the compressive and tensile strength of the concrete decreased. Bai et al. [20] investigated the influence of steel fiber amount on the mechanical properties of concrete, showing that the splitting tensile strength and flexural strength of the concrete were significantly improved by the addition of the steel fibers. However, the influence of steel fiber length on the mechanical properties of the concrete was not considered.

Although progress has been made in the reinforced theory of steel-fiber-reinforced concrete, it is significant to evaluate the influence of steel fiber amount and length on concrete performance and clarify the influence degree of the steel fiber content on the flexural strength of the concrete. In this study, simulated groups were established for concrete with different length and amount of steel fibers, and displacement–load curves were obtained from numerical simulations.

## 2. Basic Theory Extended Finite Element Method

The extended finite element method [21] is a numerical method that can effectively simulate fracture mechanics of mechanical materials. It can be used to simulate randomness and solve the crack occurrence and crack propagation processes of related path cracks without re-meshing. The XFEM function based on the conventional finite element method, which introduces an expansion function to construct a special displacement function, allows the existence of discontinuity by expanding the degrees of freedom. Then, the level set method [22] is used to obtain the actual location of a crack. On basis of this method, the extended finite element method retains advantageous characteristics of the finite element frame, such as sparsity and symmetry of the stiffness matrix.

The expansion functions include near-tip asymptotic functions to simulate the stress singularity near the crack tip, as well as discontinuous functions to represent the displacement jump at the crack surface. The displacement vector function, *u*, which uses the overall division characteristic, can be expressed by Equation (1):(1)$$u\left(x\right)=\sum_{I=1}^{N}N_{1}\left(x\right)\left[u_{1}+H\left(x\right)a_{1}+\sum_{\alpha =1}^{4}F_{\alpha }\left(x\right)b_{I}^{\alpha }\right]$$
where *N*_1_(*x*) is a commonly used node displacement function; *u*_1_ in Equation (1) represents the corresponding continuous part in the finite element displacement solution, used for all nodes in the model; *a*_1_ is the node extended degree of freedom vector, which is only valid for the element node whose shape function is cut by the crack interior; and $b_{I}^{a}$ is the node extended degree of freedom vector, valid for the element node whose shape function is cut by the crack tip; *H*(*x*) represents the discontinuous jump function along the crack surface; and *F_a_*(*x*) is the asymptotic function of stress at the crack tip. *H*(*x*) can be expressed by Equation (2):(2)$$H\left(x\right)=\left\{\begin{array}{c} 1,\left(x-x^{*}\right)n\geq 0\\ -1,\left(x-x^{*}\right)n<0 \end{array}\right\}$$
where *x* is the sample point; *x** is the nearest point in the crack spacing to *x*; *n* is the unit out normal vector of *x** in the crack. Figure 1 depicts the asymptotic function for the crack tip in an isotropic material, and it is mathematically expressed by Equation (3).
(3)$$F_{a}\left(x\right)=\left[\sqrt{r}\sin\frac{\theta }{2},\sqrt{r}\cos\frac{\theta }{2},\sqrt{r}\sin\theta \sin\frac{\theta }{2},\sqrt{r}\sin\theta \cos\frac{\theta }{2}\right]$$

In Equation (3), *r* and *θ* represent distance and angle in the polar coordinate system that take the crack tip point and crack extension direction as the pole and pole axis.

## 3. Numerical Simulation Process

Based on the finite element software ABAQUS, a four-point bending numerical model for steel-fiber-reinforced concrete was established using the XFEM module. In the simulation process, it was presumed that steel fibers are embedded in concrete, and cracking is random without prefabricated crack, regardless of boundary slip. First, randomly distributed steel fibers were generated through scripts, whose length is 10 to 15 mm. Then, a concrete model with dimensions of 120 cm × 10 cm × 10 cm was established. Concret-xfem material is used for concrete, and the maximum principal stress and fracture energy are 67.1 MPa and 4200 N/m, based on the damage evolution of energy. The steel fiber adopted Beam–Truss section property with yield strength of 235 MPa. The material parameters of the concrete and steel fibers are shown in Table 1 and Table 2 [23,24], respectively.

The steel fibers were dispersed in the concrete evenly. The static analysis is used with the nonlinearity. In addition, the field output was set including SDEG (Scalar bounties), PHILSM (Level set value Phi), STATUS, and STATUSXFEM. The historical output set the displacement and reaction force in the Z direction of the output reference point in order to obtain the load displacement curve. Tie constraints were set for all contact surfaces between rigid blocks and concrete specimens. The crack-XFEM type failure mode was set for concrete, and the XFEM-crack-growth allowing crack growth was set. The Embed Region constraint was adopted between reinforcement and concrete. We set all reinforcement to the Embed Region, and concrete specimens to the host region. Taking the concrete as the crack propagation area, the propagation characteristics were defined by a crack propagation analysis, and crack growth was allowed. For grid type, the concrete was set as a C3D8R unit with number of 12,000, and the steel fiber was set as a T3D2 unit with number of 2064. Through multiple pretreatment analyses, the sizes of the concrete grid divisions were determined and divided into 2556 units, as shown in Figure 2.

The constraint on the lower left end of the beam was set as a fixed hinge support, and it constrains displacement in the x-, y-, and z-directions and rotation in the x- and y-directions. The lower right end constraint was set as a rolling hinge support, and it constrains displacement in the y- and z-directions and rotation in the x- and y-directions (Figure 3).

To prevent local compressive break of the concrete beam, rigid body gaskets with dimensions of 10 cm × 5 mm × 2.5 mm were placed at the supports and stress points. Displacement loads were applied at two loading points, and the loads produced a vertical downward displacement of 3 mm.

The maximum principal stress criterion was adopted as the concrete cracking criterion. When the maximum stress reaches a certain value, damage occurs.

## 4. Test Results

Figure 4 presents a stress nephogram for the concrete in the experimental group under the broken load. The stresses near the fixed constraint and the load of the concrete beam are larger than that in the middle. When the tensile strength of the concrete reaches ultimate strength, the concrete crack occurs at the load-bearing portion on the left. Taking experimental groups as an example, the simulation results were shown.

### 4.1. Crack Propagation Path

The signed distance function (PHILSM) in ABAQUS is the level set function used to describe the development of the crack surface, in which the PHILSM value transitions from negative to positive unit is the location of the crack surface. Figure 5 shows the simulated crack propagation process in concrete beam at different loads.

The upper end of the concrete beam is subjected to compressive stresses and the lower end is subjected to tensile stresses. As shown in Figure 5a, the tensile strength of the concrete beam reaches the limitation firstly in the lower left area, and a crack originates, which gradually expands upward. The initial crack width is small. The steel fibers at the crack bear tensile stress, and then the total tensile strength of the concrete is improved. In Figure 5b, the concrete is in a working state with cracks. With increase in the load, the crack tip extends upward in the direction perpendicular to the tensile stress. Meanwhile, the crack width at the lower end increases, and the steel fibers are exposed. Steel fibers are broken from the bottom of the beam to the top. In Figure 5c, when the concrete enters the failure stage, the crack width increases rapidly. Most of the steel fibers at the lower end reach the limitation, and part of the steel fibers are pulled out.

### 4.2. Load–Displacement Curves for a Reference Point

Figure 6 shows load–displacement curves for the experimental groups. The first straight segment in each curve represents the elastic deformation stage of the beam, which corresponds to the state at the beginning of loading until the appearance of the crack, and the load at the end of this stage is the cracking load for the beam.

Corresponding to Figure 5b, after the cracking load, cracks appear on the inside of concrete beam, and the stiffness of the beam decreases. As the load increases gradually, the slope of the load–displacement curve begins to decrease, and the displacement increases rapidly. At the crack, most of the steel fibers embedded in the concrete are tightly bonded with the matrix, and they bear most of the tensile force.

Corresponding to Figure 5c, when the tensile force of steel fibers exceeds the bond strength limit, the load–displacement curve exhibits a second inflection point, and the stiffness of the concrete beam drops rapidly. The slope of the load–displacement curve is nearly zero, and the displacement increases sharply. This is because most of the steel fibers in the cracks are pulled out, and a small number of steel fibers are still under work. The beam enters the failure stage.

## 5. Discussion about the Mechanical Properties of the SFRC

### 5.1. Load–Displacement Curve

Figure 6a shows the influence of steel fiber length on the fracture resistance of concrete. Containing the same number of steel fibers (700), the load-bearing capacity of concrete can be continuously improved by increasing the length of steel fibers. It should be noted that the load–displacement curve improvement from the steel fiber length of 2–3 mm range to 5–10 mm range, to a 10–15 mm range is lower than that of from 10–15 mm range to 15–20 mm range where the load–displacement curve exhibits the most significant improvement. When the steel fiber length increases from a 15–20 mm range to a 20–25 mm range, there is only a slight improvement in the load–displacement curve. Therefore, different lengths of steel fiber have a different reinforcing effect.

In Figure 6b, when the number of steel fibers increases, the broken load of concrete containing same-length (10–15 mm) steel fibers increase. When the number of steel fibers increases from 100 to 300, the bearing-load increases slowly. When the number of steel fibers increases from 300 to 500, the load improvement is the largest. When the number of steel fibers increases from 500 to 700, the increase in the load is smaller again.

The breaking load for each group of the concrete was recorded, and the results were shown in Figure 7. For steel-fiber-cement-based materials, the bonding force between the steel fibers and the matrix is the key factor for preventing matrix crack propagation [25]. When cracks occur in the matrix, debonding and slippage of the steel fibers at the interface significantly influence the energy consumption (i.e., the fracture energy) during crack propagation [26].

### 5.2. Flexural Strength

#### 5.2.1. Reinforcing Effect of the Steel Fiber Length

When 2–3 mm steel fibers are added, the flexural strength of the concrete does not show significant improvement. The flexural strengths for groups B-1, B-2, B-3, and B-4 are 5.60 MPa, 5.63 MPa, 6.11 MPa, and 6.13 MPa, respectively, which are only 0.1%, 0.5%, 9.1%, and 9.8% higher than that of group A-1, respectively. Due to the short length of the steel fibers, the bonding force between the steel fiber and the matrix interface is insufficient, and the fibers are easily pulled out from the concrete during the cracking process. The tensile properties of the steel fibers cannot not fully exert, and crack propagation develops quickly. 

When steel fibers with 15–20 mm length are added, the increases in flexural strength for concrete are obvious. The flexural strength values for groups E-1, E-2, E-3, and E-4 are 5.78 MPa, 5.83 MPa, 6.87 MPa, and 7.59 MPa, respectively, which are 3.2%, 3.6%, 12.4%, and 23.4% higher than that of the A groups.

When the steel fiber lengths increase from 15–20 mm to 20–25 mm, the flexural strength of concrete does not show obvious improvement. The fiber length of the F-4 group increases by 20% with respect to that of the E-4 group, but the flexural strength increases from 14.21 MPa to 14.44 MPa, with improvement accounting for only 1.6%. This shows that the length enhancement effect of steel fibers for SFRC is insignificant from group E to group F. Increasing the length of the steel fibers cannot improve the flexural strength of the concrete. Because the tensile strength of the steel fibers is already less than the bonding force between steel fibers and matrix, the fracture resistance of the concrete cannot significantly improve.

#### 5.2.2. Reinforcing Effect of the Steel Fiber Amount

Adding an amount of steel fibers into concrete can reduce the number of cracks in the concrete matrix during the hardening and consolidation stage of the concrete. Moreover, a random distribution of steel fibers in the matrix can delay the formation and expansion of concrete micro-cracks, thereby improving the flexural strength of the concrete beam [27].

When 300 short steel fibers are added, the flexural strength improvement of concrete is very small for the experimental group. The flexural strengths for groups B-2, C-2, and D-2 are 5.63 MPa, 5.83 MPa, and 6.04 MPa, respectively, whose increases are 0.5%, 0.9%, and 2.9% related to group A-1. For long steel fibers, the improvement is very obvious. The flexural strengths for groups E-2 and F-2 are 8.65 MPa and 10.83 MPa, respectively, increasing 43.2% and 76.4% to group A-1.

When 500 steel fibers are added into concrete, the flexural strengths for groups B-3, C-3, D-3, E-3, and F-3 are 6.11 MPa, 6.87 MPa, 9.07 MPa, 11.47 MPa, and 13.75 MPa, respectively, increasing 8.5%, 17.8%, 50.1%, 32.6%, and 27.0% compared with group A-1. After the steel fiber length increases to a certain extent, increasing the amount of steel fibers can significantly increase the number of steel fibers per unit area of the concrete, making the tensile properties of the steel fiber greatly impact the concrete properties.

When 700 steel fibers are added, the flexural strengths for groups B-4, C-4, D-4, E-4, and F-4 are 6.15 MPa, 7.59 MPa, 10.19 MPa, 14.21 MPa, and 14.44 MPa, showing increases of 0.7%, 10.4%, 12.3%, 23.9%, and 5.0% compared with group A-1. Additionally, slopes of the bending strength curves become shallower again. With the increase in the steel fiber number, although the flexural strength of the concrete still increases, the growing rate of the flexural strength decreases. In fact, if the steel fiber amount continues to increase, agglomeration will also appear in the concrete.

#### 5.2.3. Optimal Steel Fiber Addition Amount

The amount and length of the steel fibers have different effects on the fracture resistance of the concrete beams. The addition of steel fiber to concrete is not only based on the mechanical strength of the SFRC, but also on the economic benefits. Therefore, a relative improvement degree evaluation index is introduced, which is analyzed depending on the length and quantity of the steel fibers. Regarding the relative improvement in the SFRC, 300 steel fibers with lengths of 20–25 mm were proposed to be an optimal addition amount and length, whose Relative Lifting Degree is 3.49 (Table 3).

### 5.3. Fracture Energy

Fracture energy (*G_F_*) is the energy required for the complete failure of concrete specimen, and it is an important index to characterize the fracture properties of materials. Its formula is expressed as:$$G_{F}=\frac{W_{0}+mg\delta_{\max}}{Ht}$$

*W*_0_ is the work by the external load; *δ*_max_ is the displacement of the specimen when failure occurs; *H* is the height of the specimen, *t* is the width of the specimen; m is the test mass between supports, and *g* is the gravitational acceleration.

Table 4 shows the change in concrete fracture energy in the experimental group when 300 steel fibers were added. It can be seen that addition of steel fibers can significantly improve the fracture energy of the concrete. Moreover, the test results show that addition of steel fiber into concrete could significantly improve the ductility and fracture energy of the concrete. However, when increasing the length of the steel fibers, the fracture energy and fracture resistance of SFRC rarely improved.

## 6. Conclusions

(1)When the steel fiber addition amount increases from 0 to 3.5%, the flexural strength of the SFRC increases from 5.60 Ma to 14.44 Ma.(2)When the length of the steel fibers increases from 2–3 mm to 20–25 mm, the flexural strength of the concrete increases accordingly.(3)Considering the economic benefits and mechanical properties comprehensively, the steel fiber amount of 1.50% in the concrete with length of 20–25 mm is proposed to be an optimal selection.(4)When the steel fiber volume content increases from 1.13% to 1.50%, the increase in fracture energy and fracture resistance for concrete is insignificant.

## Figures and Tables

**Figure 1 materials-15-07146-f001:**
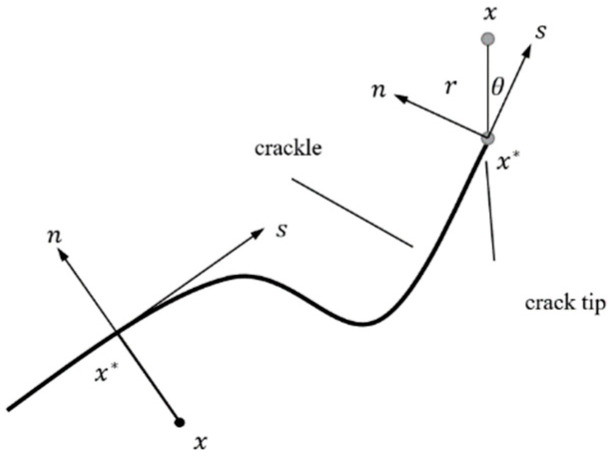
Pole axis of crack tip, where *x* is the sample point; *x** is the nearest point in the crack spacing to *x*; *n* is the unit out normal vector of *x** in the crack.

**Figure 2 materials-15-07146-f002:**
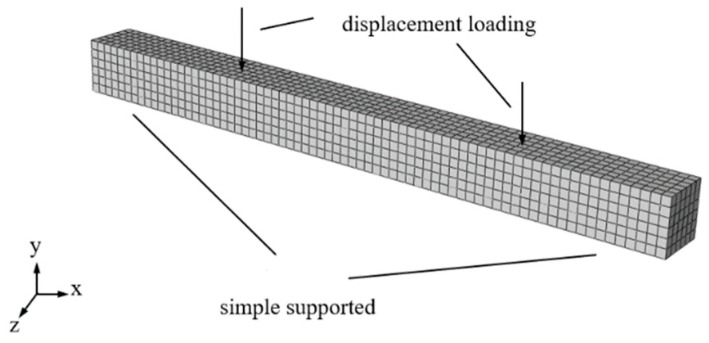
Numerical model of beam.

**Figure 3 materials-15-07146-f003:**
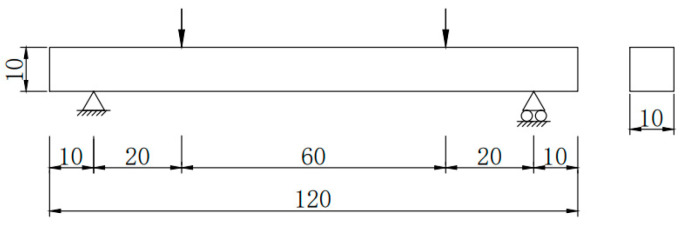
Dimension and constraints of the beam (cm).

**Figure 4 materials-15-07146-f004:**
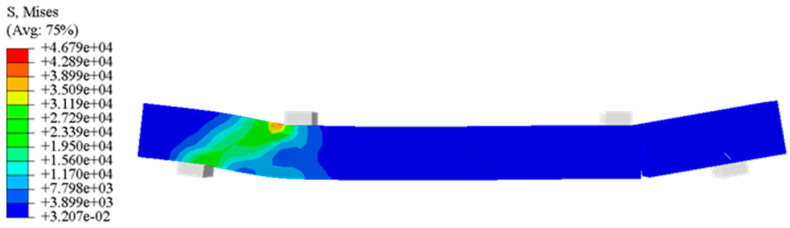
Stress nephogram for the concrete beam (N/mm^2^).

**Figure 5 materials-15-07146-f005:**
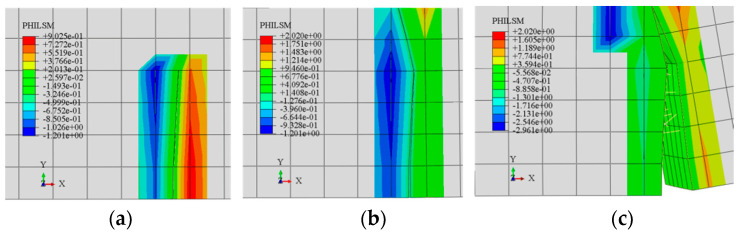
Crack propagation in beams at different loads. (**a**) 50 N; (**b**) 90 N; (**c**) 115 N.

**Figure 6 materials-15-07146-f006:**
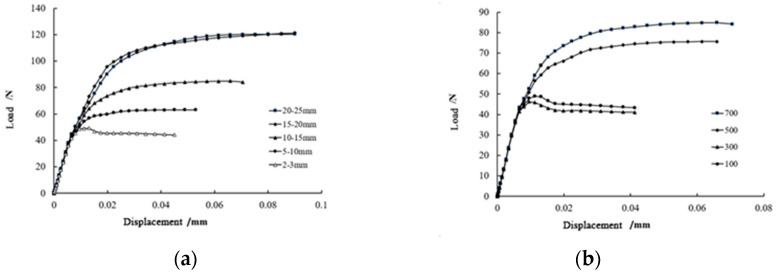
Load–displacement curves for concrete test group. (**a**) For different fiber lengths; (**b**) for different numbers of fibers.

**Figure 7 materials-15-07146-f007:**
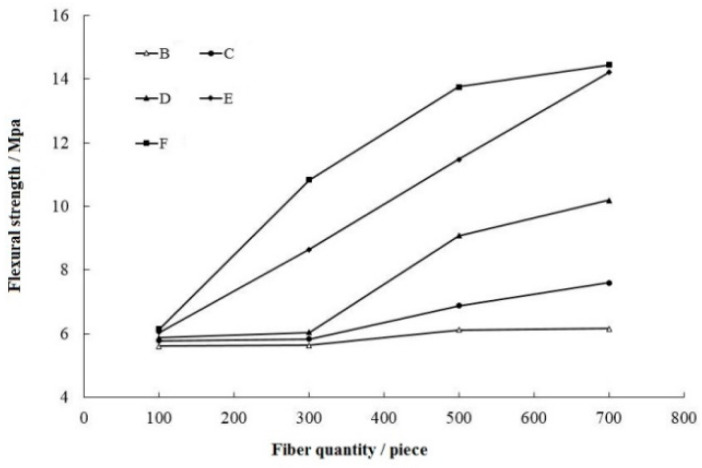
Breaking load of each experimental group.

**Table 1 materials-15-07146-t001:** Concrete material parameters [23].

Modulus of Elasticity (MPa)	Poisson’s Ratio	Maximum Principal Stress (MPa)	Fracture Energy (N·m^−1^)
36,000	0.2	67.1	4200

**Table 2 materials-15-07146-t002:** Steel fiber material parameters [24].

Modulus of Elasticity (MPa)	Poisson’s Ratio	Diameter (mm)	Density (kg·m^−3^)
201,000	0.3	0.2	7890

**Table 3 materials-15-07146-t003:** Test results of relative improvement degree evaluation index.

Test Number	Fiber Amount	Fiber Length (mm)	Fiber Volume Content (%)	Rupture Strength (MPa)	Relative Lifting Degree
A-1	0	0	0	5.60	
B-1	100	2–3	0.05	5.60	
B-2	300	2–3	0.15	5.63	0.20
B-3	500	2–3	0.25	6.11	2.04
B-4	700	2–3	0.35	6.15	1.57
C-1	100	5–10	0.13	5.78	1.38
C-2	300	5–10	0.38	5.83	0.61
C-3	500	5–10	0.63	6.87	2.02 (Continued)
C-4	700	5–10	0.88	7.59	2.26
D-1	100	10–15	0.25	5.87	1.08
D-2	300	10–15	0.75	6.04	0.59
D-3	500	10–15	1.25	9.07	2.78
D-4	700	10–15	1.75	10.19	2.62
E-1	100	15–20	0.38	6.04	1.16
E-2	300	15–20	1.13	8.65	2.70
E-3	500	15–20	1.88	11.47	3.12
E-4	700	15–20	2.63	14.21	3.27
F-1	100	20–25	0.50	6.14	1.08
F-2	300	20–25	1.50	10.83	3.49
F-3	500	20–25	2.50	13.75	3.26
F-4	700	20–25	3.50	14.44	2.53

Note: Relative improvement degree = (flexural strength of experimental group − flexural strength of control group)/(fiber volume content × 100).

**Table 4 materials-15-07146-t004:** Fracture energy.

Test Number	A	B2	C2	D2	E2	F2
Fracture Energy (N·mm^−1^)	0.12	0.56	0.87	1.23	1.71	1.76

## Data Availability

Not applicable.

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
