# Peer review of "Simulation of Four-Point Bending Fracture Test of Steel-Fiber-Reinforced Concrete"

_materials, 2022, doi:10.3390/ma15207146_

Round 1
Reviewer 1 Report
The authors of the paper used a finite element software called ABAQUS to predict the behaviour of small concrete beams reinforced with steel fibers under 4-point load. They investigated the effect of different steel fiber quatities and also the effect of fiber length on the behaviour of sets of concrete beams using the ABAQUS program. A brief literature review was carried out on the underlining theory used in the analysis of results. The Extended Finite Element Method (XFEM) which was proposed by Belytsckho and Black in 1999 is the mathematical model used by the ABAQUS software to predict the behaviour of the beams. The reviewer found three major flaws with this paper.
MAJOR FLAWS
1. THE DIMENSION OF THE TEST BEAM USED FOR ANALYSIS
In section 2 of the paper under Numerical simulations, it was stated that the size of the test beam is 120mm x 10mm x 10mm. I am asumming the dimensions stated above mean a 10mm by 10mm concrete beam section with a length of 120mm. These dimensions do not make sense at all because the authors are stating that the cross section of the beam is less than 1 inch in length and width, while they used a steel fiber length of about 25mm as a constituent material in the concrete beams.
I will like to think this is a typographical error from the authours. Probably the size of the beam dimension maybe 1800mm x 100mm x 100mm OR 180cm x 10cm x 10cm.
2. LACK OF VALIDATION OF COMPUTER SIMULATION RESULTS WITH REAL LAB EXPERIMENTAL DATA.
The finite element solutions from the ABAQUS software was not validated with lab experimental results. It would have been proper if the authors based their work on experimental data. They would have either carried out physical experiment on few of the test beams that they investigated by using the ABAQUS program. Finite element analysis for complex problem like concrete beam analysis never provides exact solution. So there is a need to compare computer results with lab experimental data. At least the load deflection result should be compared with lab experimental data
3. INSUFFICIENT INTRODUCTION AND DISCUSSION
Introduction part and results and discussions are needing for more citations and further comparison with previous studies.
MINOR FLAWS
1. “Yanlan et al. [4] studied the me-chanical properties of steel-fiber-reinforced concrete at different heating temper-atures and concluded that the tensile, compressive, and splitting strength of steel-fiber-reinforced concrete are higher than those of steel-fiber-reinforced con-crete.”
This literature review, Yanlan et al. [4] is from the introduction section. The author is trying to compare steel fiber reinforced concrete with ordinary concrete. A typo was made here. The highlighted text in red should be changed to ordinary reinforced concrete.
2. “Figure 4. Stress nephogram”
The colour plot for the stress distribution is unclear. The authors need to specify the units of the displayed results either in N/m2, N/mm2 or MPa, GPa.
3. “Figure 5. Crack propagation in beams at different loads. (a) 50 N load; (b) 90 N load; (c) 115 N load.”
The color plot for the crack propagation in the beam at different loading is unclear. The author needs to show the crack progression in relation to the full size of the beam.
Author Response
Thanks to reviewer for your constructive and detailed comments. We have carefully revised the article according to the comments. The detailed amendments and replies are as follows:
1、The dimension of the test beam used for analysis. In section 2, it was stated that the size of the test beam is 120mm × 10mm × These dimensions do not make sense at all because the authors are stating that the cross section of the beam is less than 1 inch in length and width, while they used a steel fiber length of about 25mm as a constituent material in the concrete beams.
Reply: We are very sorry for this slip. The dimension of the concrete beam is 120cm × 10cm × 10cm. We correct this mistake in the revision.
2、Lack of validation of computer simulation results with real lab experimental data.
Reply: We absolutely agree with you, and your suggestion is of great practical significance. We would like to carry out such tests, but we can't make up the work in this short time. In future work, we plan to carry out this kind of tests.
3、Insufficient introduction and discussion. Introduction part and results and discussions are needing for more citations and further comparison with previous studies.
Reply: We rewrite a part of manuscript. We really try our best to improve the quality of the article. We have revised 788 points in the revision. Please see the revised article for more details.
4、This literature review, Yanlan et al. [4] is from the introduction section. The author is trying to compare steel fiber reinforced concrete with ordinary concrete. A typo was made here. The highlighted text in red should be changed to ordinary reinforced concrete.
Reply: We change this error together with many other errors.
5、Figure 4. Stress nephogram” The colour plot for the stress distribution is unclear. The authors need to specify the units of the displayed results either in N/m2, N/mm2 or MPa, GPa.
Reply: The unit is N/mm2, we add it in the paper.
6、The color plot for the crack propagation in the beam at different loading is unclear. The author needs to show the crack progression in relation to the full size of the beam.
Reply: we replace it with a clearer picture.
Reviewer 2 Report
The present manuscript is interesting and well-organised. The topic is in line with the object of the Journal.
However, in the reviewer's opinion, some revisions are needed in order to meet the quality requested for publication on Materials, as highlighted in the marked-up file in attachement.

Author Response
Thanks to reviewer for your constructive and detailed comments. We have carefully revised the article according to the comments. The detailed amendments and replies are as follows:
1、Generally, concrete is assumed as a material characterised by a brittle or quasi-brittle behaviour. Thereofere, in the reviewer's opinion, plasticity is not a peculiarity of such a material.
Reply: Accepted and corrected.
2、Please, significantly increase the number of literature works referenced here. Some works to be cited are reposrted in the following. Please also include in the text the following sentences.
Reply: Accepted and corrected.
3、Please, significantly increase the number of literature works referenced here.
Reply: Accepted and corrected.
4、Insert at least one reference here.
Reply: Accepted and corrected.
The reviewer gives a lot of useful suggestion, and we accepted almost all the suggestions. Although it may not be perfect, we've made a lot of substantial changes indeed.
Thank you again for your helpful and constructive suggestion.
Reviewer 3 Report
The following comments are drawn:
- The authors state that the randomly distributed steel fibers were generated through scripts. Does the randomly distribution of fibers account for the wall-effect that is known to occur in practice? If so, how was this implemented in scripts?
- The interaction between the steel fibers and the cementitious matrix governs the mechanism of SFRC. How was the adherence between fiber and matrix considered in the model? Was the shear interaction considered in the ABAQUS model?
- The authors opted for exploring the influence of length and number of fibers. The reviewer suggests that the authors evaluate the influence of the aspect ratio, which may be more representative and does not induce a biased way of thinking about the use of fibers.
- The load displacement curves presented in Fig. 6 does not properly represent the behavior of SFRC. Given that the tensile strength of the matrix is constant, which is controlled by the maximum principal stress criteria, the increase in fiber content changes the post-crack behavior of concrete - which can be a strain-softening or strain-hardening. The behavior evidenced in Fig.6 shows basically a perfect elasto-plastic material, which is not the case for SFRC.
- The matrix tensile strength (i.e. crack strength) of concrete does not change significantly with the increase in fiber content. The fiber-matrix interaction begins to be significant after the matrix cracks. The results obtained through the numerical simulation shows that the flexural strength greatly increases, which does not represent the actual behavior of SFRC. The reviewer suggests that the authors verify the model in order to properly represent the mechanics of SFRC.
- The simulation of SFRC is considerably more challenging than conventionally reinforced concrete. The fiber-matrix interface plays a significant role in the behavior of the material. Therefore, in case of discrete-explicit numerical models, this interaction must be considered. Alternatively, sectional models may be employed if the constitutive equation of the SFRC is provided - especially under tension. The reviewer suggests that the authors verify the constitutive equations provided in fib Model Code-10.
- The numerical results must be compared and validated to experimental results. Given the results presented by the authors, the reviewer is concerned because the ABAQUS model does not properly represented the behavior of SFRC under flexural loading. This is specially true because the model considered a perfect bond between fiber and matrix, which does not represent the bond-slip of the steel fibers and the strain-softening or strain-hardening behavior.
Therefore, the reviewer recommends a considerable review of the model considering a more appropriate approach to simulate the complex behavior of SFRC.
Author Response
Dear reviewer, thank you for your valuable professional advice. We very, very agree with you. What you have said is luminous. The force analysis of SFRC is a complex behavior. I would like to divide this complex problem into several parts, and then take out one of the most important influencing factors for analysis. I consider that the amount and shape of steel fibers have the greatest influence on the performance of concrete, so we evaluate this influence here. Therefore, we chose to ignore the wall-effect and shear interaction. We are also willing to conduct experiments in the future to verify the simulation results. I like digital simulation, but as a beginner, it's hard for me to completely re-model in the short term. Your advice has given me a lot of inspiration. In the future work, I will gradually improve this simulation. Again, I would like to express my sincere thanks for your professional and rigorous opinion.
Reviewer 4 Report
Dear authors.
A job well done. But there are major problems.
-At the end of the article, do not use the word" 5. Summary", but use "5. conclusion".
-What is the "Table 1. Concrete material parameters" reference? please add.
-Why was this study not done using the finite element software ATENA 3D, which is special for the numerical simulation of the behavior of reinforced concrete structures in three dimensions?
-The results of several types of research in the field of modeled beam samples using steel fibers and comparing them with concrete beams without fibers show that fiber concrete increases the total energy absorbed before complete rupture. What is the innovation of this article?
-In this research, how are the results of the numerical model of fiber concrete beam verified with the results of laboratory data? The description is little.
-The number of previously reviewed references and studies is very few and some are old, so more and more recent references should be used.
Thanks.
Author Response
Thanks to reviewer for your constructive and detailed comments. We have carefully revised the article according to the comments. The detailed amendments and replies are as follows:
1、At the end of the article, do not use the word" 5. Summary", but use "5. conclusion".
Reply: Accept and correct.
2、What is the "Table 1. Concrete material parameters" reference? please add.
Reply: added.
3、Why was this study not done using the finite element software ATENA 3D, which is special for the numerical simulation of the behavior of reinforced concrete structures in three dimensions?
Reply: Our lab is planning to buy this software, and to my shame, we can't use it yet.
4、The results of several types of research in the field of modeled beam samples using steel fibers and comparing them with concrete beams without fibers show that fiber concrete increases the total energy absorbed before complete rupture. What is the innovation of this article?
Reply: We would like to give a detailed reference to the effect of the amount and length of steel fiber on the flexural capacity of the concrete.
5、In this research, how are the results of the numerical model of fiber concrete beam verified with the results of laboratory data? The description is little.
Reply: We want to do this validation trial, but we haven't done it yet because of time constraints.
6、The number of previously reviewed references and studies is very few and some are old, so more and more recent references should be used.
Reply: We added a lot of literature and revisions. And revisions are as much as 788 in the revised paper. We are trying to improve the quality of the articles as hard as we can.
Thank you again for your helpful and constructive comments.
Round 2
Reviewer 1 Report
The authors of the paper used a finite element software called ABAQUS to predict the behaviour of small concrete beams reinforced with steel fibers under 4-point load. They investigated the effect of different steel fiber quatities and also the effect of fiber length on the behaviour of sets of concrete beams using the ABAQUS program. A brief literature review was carried out on the underlining theory used in the analysis of results. The Extended Finite Element Method(XFEM) which was proposed by Belytsckho and Black in 1999 is the mathematical model used by the ABAQUS software to predict the behaviour of the beams. While revıwıng thıs paper for the second tıme, it was found that the authors made corrections and provided more information about the issues mentioned in my first review. These corrections include the following:
1. The dimension of the test beam was corrected with reasonable dimension.
2. Still the numerical analysis was not validated with experimental lab data. There is no way of determining the degree of accuracy of the theoritical model used for their research.
3. The literature reviw of Yanlan et al. [4] was corrected as I advised.
4. Unit of measurement for the color plot in figure 4 was provided in the revised paper
5. Although the full length of beam was not depicted to show crack progression in Figure 5, the authors provided more explanation of the crack progression in their revised paper.
Overall, I think this paper is just adequate for publication in your journal.
Author Response
Thank you for your help and comments. Your comments greatly improve the quality of the article.
Reviewer 2 Report
The manuscript has been significantly revised and improved. Therefore, in the reviewer's opinion it can be now accepted for publication on Materials.
Author Response
Thank you for your kind help and constructive comments.
Reviewer 3 Report
Dear Authors,
The work needs to be properly reviewed and the study must be adequately conducted, especially given the limitations pointed during the review process and the approach that is not representative of the actual behavior of SFRC.
Author Response
Thank the reviewer for his professional and constructive comments. These comments greatly improve the quality of this paper.
1 The authors state that the randomly distributed steel fibers were generated through scripts. Does the randomly distribution of fibers account for the wall-effect that is known to occur in practice? If so, how was this implemented in scripts?
Reply: Thank the experts for their valuable advice. This paper focuses on the influence of random distribution of steel fibers on the bending and fracture performance of concrete, the main concern is the realization of random distribution of steel fibers in the finite element simulation model, and the Embed region constraint between steel fibers and cement can simulate the steel fibers from the cement.
2 The interaction between the steel fibers and the cement matrix governs the mechanism of SFRC. How was the adherence between fiber and matrix considered in the model? Was the shear interaction considered in the ABAQUS model?
Reply: EMBED constraint is adopted between fiber and matrix, which is equivalent to setting the same translational freedom of fiber node and matrix node. In addition, the current study does not consider shear interaction, because it is not the main form of steel fiber failure. However, it will be studied in the future considering its influence on the mechanical properties of steel fiber reinforced concrete.
3 The authors opted for exploring the influence of length and number of fibers. The reviewer suggests that the authors evaluate the influence of the aspect ratio, which may be more representative and does not induce a biased way of thinking about the use of fibers.
Reply: Thank the experts for their valuable advice. In this paper, when the influence of fiber length is studied, the fiber diameter is fixed, so the fiber length can represent the aspect ratio of the fiber. The aspect ratio of fiber will be used as the variable in subsequent studies.
4 The load displacement curves presented in Fig. 6 does not properly represent the behavior of SFRC. Given that the tensile strength of the matrix is constant, which is controlled by the maximum principal stress criteria, the increase in fiber content changes the post-crack behavior of concrete - which can be a strain-softening or strain-hardening. The behavior evidenced in Fig.6 shows basically a perfect elasto-plastic material, which is not the case for SFRC.
Reply: Thank you for your question. The author focuses on the maximum bearing capacity of steel fiber reinforced concrete (SFRC) in four-point bending experiment considering factors such as fiber pulling out and matrix cracking, which can be determined by the load displacement curve. In addition, more detailed finite element model establishment and simulation will be carried out later.
5 The matrix tensile strength (i.e. crack strength) of concrete does not change significantly with the increase in fiber content. The fiber-matrix interaction begins to be significant after the matrix cracks. The results obtained through the numerical simulation shows that the flexural strength greatly increases, which does not represent the actual behavior of SFRC. The reviewer suggests that the authors verify the model in order to properly represent the mechanics of SFRC.
Reply: The question raised by the experts is very professional. Due to the interaction between the fiber and the matrix in the finite element model, the calculation results preliminarily show that the addition of steel fiber does improve the flexural strength of concrete. As for its actual mechanical properties, further experimental studies will be carried out.
6 The simulation of SFRC is considerably more challenging than conventionally reinforced concrete. The fiber-matrix interface plays a significant role in the behavior of the material. Therefore, in case of discrete-explicit numerical models, this interaction must be considered. Alternatively, sectional models may be employed if the constitutive equation of the SFRC is provided - especially under tension. The reviewer suggests that the authors verify the constitutive equations provided in fib Model Code-10.
Reply: Thanks to the expert's suggestion, the steel fiber adopts the cross section model, and its constitutive equation uses the classical elastic-plastic constitutive equation.
7 The numerical results must be compared and validated to experimental results. Given the results presented by the authors, the reviewer is concerned because the ABAQUS model does not properly represented the behavior of SFRC under flexural loading. This is specially true because the model considered a perfect bond between fiber and matrix, which does not represent the bond-slip of the steel fibers and the strain-softening or strain-hardening behavior.
Reply: Thank you for your advice. The binding slip and strain softening or strain hardening behavior of steel fibers will be taken into account in the subsequent finite element simulation analysis. In addition, for the problems that experts are concerned about, the subsequent research will add experimental and simulation results for comparison.
Reviewer 4 Report
thanks
Author Response
Give my sincere thanks to the reviewer for his help and constructive comments.